# Probing the Use of Silane-Grafted Fumed Silica Nanoparticles to Produce Stable Transformer Oil-Based Nanofluids

**DOI:** 10.3390/ma14247649

**Published:** 2021-12-12

**Authors:** Muhammad I. Qureshi, Basit Qureshi

**Affiliations:** 1Department of Electrical Engineering, Lahore University of Management Sciences, Lahore 54792, Pakistan; mqureshi287@gmail.com; 2Department of Computer Science, Prince Sultan University, Riyadh 11586, Saudi Arabia

**Keywords:** fumed nano-silica, plasma synthesis, surface modification, dispersibility, plasma reactor, Weibull statistical analysis, polar/unpolar silanes

## Abstract

In this experimental investigation, hydrophobic silane-grafted fumed nano-silica was employed in transformer oil to formulate nanofluids (NFs). A cold-air atmosphere-pressure plasma reactor working on the principle of dielectric barrier discharge was designed and utilized to functionalize the surface of these nanoparticles. A field emission scanning electron microscope (FE-SEM) coupled with energy-dispersive X-ray (EDX) module and Fourier transform infrared (FTIR) spectroscopy were used to scan surface features of new and plasma-treated nanoparticles. The study revealed considerable changes in the surface chemistry of nanoparticles, which led to good dispersibility and stability of nanofluids. The measurements of AC breakdown voltages (AC-BDV) of nanofluids so prepared were conducted according to IEC-Std 60156, and a significant improvement in the dielectric strength was achieved. A statistical analysis of these results was performed using Weibull probabilistic law. At a 5% probability of failure, modified nanofluid remarkably exhibited a 60% increase in breakdown voltage. The dielectric properties such as variation of $\varepsilon_{r}$ and tan δ in temperature of up to 70 °C were measured and compared with untreated fluid. Results exhibit an increase in tan δ and a slight decrease in permittivity of nanofluids. The analysis also revealed that while unpolar silane coating of NPs increased the breakdown strength, the polar-amino-silane-coated NPs in oil resulted in a drastic reduction. Details of this antagonistic trend are elaborated in this paper.

## 1. Introduction

The advent of nanotechnology has opened new horizons in the development of nanofluids (NFs). NFs have successfully been applied to formulate transformer oil, which is the bloodline of the power transformers that act as a backbone of an electric power network. These NFs have gained the attention of researchers worldwide due to the enhancement of their insulating properties and much better cooling performance. As a result, these efforts have helped downsize the electrical apparatus with large capacity, lower weight, and high reliability.

Choi et al. [1] put forward the term “nanofluids” (NFs), which refers to the possibility of elevating the heat transfer coefficient by the addition of nanoparticles in various base fluids, including mineral insulating oil (MO). Dispersion of these nanoparticles in NFs so prepared to augment thermal efficiencies has gained much attention among researchers in the recent past [2]. The main reason for the heat transfer enhancement of NFs is that the suspended NPs increase the thermal conductivity of the fluids, and the chaotic movement of ultrafine particles increases the fluctuation and turbulence of the fluids, which accelerates the energy exchange process [3]. Nanoparticles are mostly constructed from metallic elements (Al, Cu), inorganic oxides (Al_2_O_3_), nitrides (AlN, SiN), or non-metallic elements (graphite, graphene, carbon nanotubes), etc. More recently, “green” nanoparticles derived from plants and seeds have also been synthesized to formulate eco-friendly, stable, green nanofluids for increased thermal properties [2,3]. These have now emerged as an alternate research area that is growing with a great deal of interest. The cooling efficiency of nanofluids depends mostly on the volume fraction of nanoparticles, their size, shape, and interface between nanoparticles and the host liquid [4]. In this context, investigators have synthesized nanoparticles by grafting specific functional moieties on their surfaces [5,6,7,8]. de Almeida et al. successfully fabricated multicore and core-shell silica-coated maghemite nanoparticles of different sizes and achieved, in nanofluids so formulated, excellent zeta potential, leading to good dispersibility and colloidal stability for use in several base fluids including transformer insulating oils [9]. Ionic liquid-based nanofluids are yet another area of current research for heat transfer fluids [10]. Thermal properties of nanofluids are an extensive research area that is addressed in the past few decades for their perspective technological applications, such as in heat exchangers [11], manufacturing systems [12], oil and gas [13], automobile industry [9], nuclear power plants, space stations, air crafts, solar panels [14,15] and more recently in microbial fuel cell technologies [16].

It was also shown that the AC dielectric strength of insulating oil-based nanofluids can also be significantly improved by adding some specific NPs. Segal et al [17] are the first researchers to study the modification of magnetic (Fe_3_O_4_) nanoparticles (NPs). They reported an increase in AC-BDV two times higher than MO. Generally, three common types of NPs have been widely used to develop NFs, such as conductive, semi-conductive, and insulating. The most frequently investigated NPs are zinc oxide (ZnO), copper oxide (CuO), fullerene (C60), and aluminum nitride (AlN) [18]. Dhar et al [5] observed that the addition of traces of carbon nanotubes (CNT) or graphene improves the AC-BDV of MO by about 70–80%. Besides the reported results that focus on the singular impact of NPs, more recently some investigators studied the impact of mixtures of NPs on the dielectric strength of nanofluids so prepared. Beroual and Khalid [19] used mixtures of (Fe_3_O_4_ + SiO_2_) and (Fe_3_O_4_ + Al_2_O_3_) NPs and reported almost 90% enhancement in the average values of AC-BDV. Thabet et al. [20] studied the electrical breakdown of multi-nanoparticles (ZnO, TiO_2_, LiTaO_3_, Fe_3_O_4_, MgO, SiO_2_, and graphite), and showed that multi-nanoparticles are more efficient than individual nanoparticles. However, both the single- and multi-nanoparticle-formulated NFs exhibited improved AC-BDV until a critical loading of NPs in the hot oil but decreased gradually with a further increase in NP concentration in the oil. This problem occurs due to agglomeration and sedimentation in the NF. Thus, the most important matter in nanofluids is to gain its stability for longer periods. One of the ways to cope with this problem is the use of chemical surfactants. So, a variety of surfactants have been applied by investigators, since they do aid in the efficient dispersal of NPs, but hereto, there are conflicting results in the reported data regarding the optimum preparation method of NFs, which can play a role in avoiding agglomeration and instability of NFs [21,22].

During the past decade, yet another technique reported in the literature is the treatment of NPs by cold atmospheric-pressure plasma before mixing them in the host oil, which results in achieving stabilized nanofluids [23]. More recently, He et al. [24] have modified fumed silica with plasma and showed that it uniformly disperses in host insulation and affects the charge-trapping density and decreases the amount of injected charge. Unlike wet chemical surfactants, plasma treatment does not generate waste, and is a fast and energetically advantageous process. The carrier gas can be Ar, O_2_, H_2_, C_2_H_2_, or simply air. Depending on the type of gas, plasma treatment can be used for imparting certain functional groups via materials’ reactions with ionized gas particles. Interestingly, all types of solid surfaces can be treated with plasma.

Recent studies have not only shown that grafting amino or vinyl silane coupling agents on the surface of silica NPs can effectively control the effects of agglomeration and dispersion, but they also improve the physico-chemical properties of insulating material [6,25]. In the case of solid polymers and epoxies that form important insulating materials for use in Ultra High Voltage (UHV) apparatus, amino-based silane coupling agents with different chain lengths have been successfully utilized to graft SiO_2_ NPs, which resulted in a considerable increase in the properties of nanocomposites so formulated [6,25,26]. However, similar studies using silane-grafted NPs modified under cold-air plasma discharge to prepare NFs have not yet been much explored.

In this experimental study, we have compared the performance of transformer oil (MO) with the addition of hexamethyldisilazane (HMDS)-grafted hydrophobic fumed silica nanoparticles, as well as when these NPs are functionalized under cold-air atmospheric-pressure plasma discharge. The surface chemistry of NPs was evaluated using the FTIR-spectroscopic technique and FE-SEM coupled with energy-dispersive X-ray (EDX) spectroscopy. Properties of these NFs such as AC-BDV, permittivity, dielectric loss (tan δ), as well as their long-term stability are also evaluated. These results are also compared with the amino-silane-coated hydrophilic SiO_2_ NPs-based transformer nanofluids reported in the literature [27], which display a drastic reduction in AC-BDV.

This work demonstrates that cold atmospheric-pressure plasma treatment of NPs changed their surface chemistry, producing stronger chemical bonds. These resulted in excellent dispersibility and stability of the oil-based suspension. As a result, a significant improvement in dielectric strength was achieved. At a 5% probability of failure, this modified nanofluid exhibited a 60% increase in breakdown voltage, which is attributed to favorable space-charge trapping and de-trapping velocity in the oil. Furthermore, we observe that the relative increase in conductivity of oil as compared to base oil is due to the electric double layer (EDL). The relative permittivity decreased with temperature but was not much different from base oil in the temperature range up to 70 °C. Although plasma treatment of HMDS treated silica increased the dielectric properties of the oil, their treatment with amino-polar silane considerably reduces their dielectric integrity. The reason behind this antagonistic trend is delineated in this paper.

## 2. Experimental Methods

The mineral oil-based nanofluid (MO-N) reported here was prepared by using fumed SiO_2_ NPs obtained from NOVIK Degussa GmbH, with the trade name Aerosil-R812. R812 is pyrogenic silica after being treated with hexamethyldisilazane (HMDS), with high specific surface area and profound hydrophobicity. Table 1 illustrates the characteristics of this salinated fumed silica as supplied by the manufacturer [28], whereas Figure 1 depicts the molecular structure of HMDS.

The mineral oil (MO) used was obtained from a newly manufactured 36 kV-rated power transformer and was filtered through a 16 μm pore-size sintered glass filter and dehydrated before the preparation of NF formulations. In this case, the oil sample was dried with silica-gel pallets (with a diameter of 4–6 mm) at a ratio of 1.0 g of silica to 100 mL of oil under constant stirring of 600 rpm for 4 h. The moisture content of this oil sample used was set at 15 ppm and was determined by the Karl Fisher titration method according to ASTM-Std.-D 6304.

### 2.1. Plasma Reactor

A small-scale plasma reactor based on the principle of dielectric barrier discharge (DBD) was designed and implemented. Figure 2a shows its outline. Cold-air atmosphere-pressure plasma discharge was generated by a 50 Hz power supply maintained at an output of 20 kVrms. The aluminum high-voltage disc electrode (see Figure 2b) was embedded with an array of carbon steel gramophone needles, each placed 7–8 mm apart since this inter-needle gap has been reported to produce optimum discharge magnitude in the stressed electrode gap [29]. The tip radius of each needle was 60 μm while its projected length was 10 mm. A 2 mm thick quartz glass plate was placed on the top of the ground electrode to form DBD. The body of the chamber was made of a Perspex cylinder. It worked in the open air in a temperature-controlled laboratory where the ambient temperature was maintained around 25 °C and ~30% relative humidity. One of the advantages of this study was preparing the process in a simpler (air) environment and without the need for vacuity equipment. This study was used to direct plasma processing in which ions, together with the electrons, have a great role in making reactions. More details about the electrical/electronic measurement techniques and parameters investigated to achieve optimum results are presented elsewhere [30].

In each experiment, we poured around 30 mg NPs into a Petri dish using a digital balance with measuring accuracy of 0.1 g and placed it on the glass plate. A gap of 5 mm was kept between the tip of the high voltage electrode needles and the surface of NPs. For this purpose, a precise 5 mm thick plastic spacer was placed between the top surface of the Petri dish and the needle electrodes. The gap was adjusted using the lifting/lowering mechanism system, as shown in Figure 2a. This formation resulted in a plasma discharge projected at the surface of NPs. After treatment of 5 min, the supply was switched off and the NPs were stirred with a glass rod so that all NPs were exposed to plasma discharge uniformly. This procedure was repeated six times. Field emission scanning electron microscope (FE-SEM) images, shown in Figure 3, compare the untreated and plasma-treated NPs. The untreated particles are in the form of agglomerates while the plasma-treated particles are loosened and mostly separated from each other. This is due to the impact of filamentary micro discharges that impinge on the surface of NPs causing a change in their surface chemistry, as illustrated later in Section 3.1. It is also reported in the literature that under the impact of plasma discharges, the loose hydrocarbon layer deposited on the silica surface, if any, is blown off and thus after modification prevents re-aggregation of the plasma modified silica [25].

### 2.2. Characterization of Nanoparticles

FE-SEM (JEOL JSM6360A, Tokyo, Japan, equipped with an EDX module) was used to determine surface morphology and elemental analysis of as such and plasma-treated silica coatings. The microscope was operated at 5 kV with a working distance of 4.5 mm. The functional group identification was studied by FTIR spectroscopy using a Thermo Scientific Nicolet iNIO FTIR microscope housing a Germanium micro-tip accessory operated in the Attenuated Total Reflectance (ATR) mode. The samples were scanned in the range 4000–400 cm^−1^ at a spectral resolution of 1 cm^−1^.

### 2.3. Nanofluid Preparation

Two types of nanofluids, A and B, were prepared using a two-step process as delineated in the flow chart of Figure 4. For Sample A, around 300 mL of oil sample was poured into a beaker and was used for each set of experiments. In the first sample, untreated NPs in a concentration of 0.03 g/L were added and were stirred using a magnetic stirrer for 30 min. This beaker was then transferred to the water bath of the sonicator and subjected to acoustic wave energy for 15 min to break down any agglomerates in the sample. Later this sample was placed in a vacuum chamber maintained at 0.01 MPa for 24 h. This step was necessary to get rid of any microbubbles produced as a result of ultrasonication. This sample is denoted here as (MO-N). For Sample B, the same procedure was adopted, except in this case, plasma-treated NPs were used in a concentrate of 0.03 g/L and mixed in 300 mL of oil. The above procedure was repeated. This sample is denoted here as (MO-NP).

### 2.4. Measurement of Breakdown Voltage

The breakdown voltage (BDV) was measured using a Foster type 90 oil tester containing brass mushroom-faced electrodes. They were spaced initially at 2.50 ± 0.05 mm apart, but the gap was later readjusted to 1.0 ± 0.05 mm due to higher BDV of samples that exceeded the upper limit of the test set. The 50 Hz AC voltage was increased progressively at a rising rate of 2.0 kV/s, as per IEC-60165. A set of six breakdowns was compiled and average BDV was deduced from five consecutive sets (i.e., 30 measurements in total for a sample of oil). The purpose of compiling 30 measurements is to perform statistical analysis. It has been shown earlier that BDV of the MO sample fits better on 2-parameter Weibull statistical distribution [30,31]. These measurements were conducted at room temperature (25 °C). This procedure was repeated for each sample investigated here.

### 2.5. Measurement of Dielectric Constant ($\varepsilon_{r}$) and Dissipation Factor (tan δ)

The dissipation factor is related to the measure of power dissipated in the transformer fluid. A low value indicates the minimum power dissipated, while a higher value indicates the presence of contamination. The values of tan δ and $\varepsilon_{r}$ in transformer oil are very important and are evaluated before being utilized. The controlling parameters of these are presented in Equations (1)–(3) [32]. The power loss (PL) defines the specific dielectric energy loss. Both are expressed as:(1)$$\tan\;\delta =\frac{\varepsilon^{\prime }}{\varepsilon_{r}}$$
(2)$$P_{L}=\omega \;E^{2}\varepsilon_{o}\varepsilon_{r}\tan\delta$$

Here *E*, is the applied electric field (Vm^−1^), $\varepsilon_{r}$ is the real dielectric permittivity $\varepsilon^{\prime }$ is imaginary permittivity, $\varepsilon_{o}$ is vacuum permittivity and ω is the angular frequency. $\varepsilon_{r}$ and tan δ of tested NFs were measured using Tettex Schering bridge type 2811 and Test cell type 2903, following IEC-std 60247. The test system was equipped with a temperature regulator, which controls the temperature within a temperature accuracy of ±1 °C. The three-terminal test cell was made of a stainless-steel body and the values of $\varepsilon_{r}$ were measured as follows. The ratio of the capacitance of test cell filled with oil to the capacitance of empty cell is given as:(3)$$\varepsilon_{r}=\frac{C_{x}}{C_{o}}$$
where *C_x_* is the capacitance of an oil-filled cell and *C_o_*, the capacitance of an empty cell. The value of tan δ was obtained under a voltage of 1500 volts using Schering bridge with an accuracy of 10^−4^.

## 3. Results and Discussion

### 3.1. Surface Analysis of Nanoparticles

Figure 5 shows the FTIR spectra of (a) Silane-grafted fumed silica NPs and (b) The same after plasma treatment. In Figure 5a, the absorption bands at 3420 cm^−1^ and 1629 cm^−1^ are attributed to the silanol group and adsorbed water respectively. The absorption band at 2965 cm^−1^ corresponds to stretching vibrations of CH_3_. The absorption peak at 1018 cm^−1^ indicates the stretching vibration of Si-O-Si [26,33]. The absorption band at 3420 cm^−1^ is typical of hydrogen-bonded (O-H) groups called hydroxyl groups, which are content attached to the surface of NPs [7,8]. Usually, in untreated silica this band has higher intensity, but due to sialylation, its peak is reduced. With a thicker layer of HMDS, this peak tends to almost disappear, showing that the NP surface has become dry. This effect is clear in Figure 5a where the intensity of the hydroxyl band has shrunk to a smaller peak and -OH radicals are substituted with -CH_3_ at 2965 cm^−1^, rendering the surface of NPs hydrophobic [33].

In the case of plasma-treated NPs, as depicted in Figure 5b, the (O-H) band at 3420 cm^−1^ has widened such that it has overlapped the -CH_3_ peak at 2965 cm^−1^, while its intensity has also increased considerably as compared to that of the control sample. The plasma discharge in the air that impinged the NP surface involved many active species such as, electrons, ions, and photons. The bombardment of these active species resulted in the formation of radicals. The recombination of these radicals resulted in modifications of NPs surface morphology. In addition, atoms of active species reacted with oxygen-containing functional groups further releasing oxygen atoms. Thus, the Si-O, Si-C bonds are also broken down. As a result, the surface silane groups at 1018 cm^−1^ are broken down resulting in a decrease in the intensity of the Si-O-Si group as shown in Figure 5b. The breaking of such silane groups due to energetic ions can further create activated surface sites to form chemical bonds with the surrounding host matrix after these NPs are added to the transformer fluid. This change is further confirmed from EDX analysis.

Table 2 shows the quantitative atomic percent concentration and ratio of (O/Si) after plasma treatment of NPs, as obtained through EDX analysis. It is observed that after plasma treatment, the Si content increases while the O content decreases. The O content on the surface decreased from 75% to 68% on the plasma-treated NPs. Similarly, the (O/Si) ratio of plasma-treated NPs decreased from 3.12 for untreated to 2.12 for plasma-treated surfaces. Thus, the plasma can generate a wide range of active species, which can change the surface chemistry of the HMDS grafted silane silica.

### 3.2. Dielectric Properties: Tan δ and $\varepsilon_{r}$ Measurement

Tan δ is an important test for transformer fluids as well as NFs since it is an important parameter for investigating the health of transformer insulation. Therefore, it is measured at power frequency (50/60 Hz) as well as at various frequencies, with dielectric spectroscopy as a parameter related to the dielectric loss (P_L_) in liquid to determine the deterioration level of oil insulation properties. In this investigation, tan δ of MO and nanofluid (MO-N) was measured as a function of temperature.

Figure 6 shows that tan δ increases with temperature in the measured range of up to 70 °C. The increase in tan δ is almost 30% more at 70 °C than in untreated MO. This is consistent with the results reported by Dong et al. [25] for NFs prepared with Aluminum Nitride (AlN) nanoparticles mixed in transformer oil. This is due to an increase in the conductivity of the colloid. According to the theory of colloid and surface chemistry, there exists an electrical double layer (EDL) around each NP surface. The surface charge of NPs, together with the ion cloud that constitutes EDL, contributes to the enhancement of conduction through the electrophoretic transactions. When the NP volume fraction is low, as in the present experiment, the nanofluid can be treated as a monodisperse system. The mobility of charged NPs caused by the Brownian motion increases with the increase of temperature [34,35]. It is not much sensitive to the conductivity of the host liquid but is more sensitive to the formation of EDL [7,8].

Figure 7 illustrates the variation of $\varepsilon_{r}$ as a function of temperature up to 70 °C. We have investigated this temperature range since the power transformers mostly operates in this temperature range. It is clear that $\varepsilon_{r}$ decreases almost monotonically with the rise in temperature in NF and exhibits lower values of $\varepsilon_{r}$ than in pure MO. Although the trend of decrease in $\varepsilon_{r}$ in NF is consistent with the results reported by Jin Mio et al. [36] for NF prepared using ZnO NPs in MO but is opposite to increase in $\varepsilon_{r}$ reported in their experiments and attributed it to NPs inner polarization. However, we believe that it is due to the lower conductivity of the silane-capped fumed silica. Voorthuzyen et al. [37] reported that by using silane agents such as HMDS, the surface conductivity of silica is reduced by at least a factor of 1000. Although lower $\varepsilon_{r}$ in silica-based NF may be useful toward the lowering of dielectric loss in transformer, it may pose a minor problem in the design of paper–liquid composite, as the $\varepsilon_{r}$ of Kraft paper used in transformer windings is ~4. Thus, more electrical stress will become concentrated in the oil. However, the enhanced dielectric strength of NF, as elaborated in Table 3, will overcome this deficiency.

### 3.3. AC Breakdown Voltage

Power utilities measure average values of AC breakdown strength of transformer oil to check the health of liquid insulation. However, the transformer design engineers have to rely on the minimum withstand of the level of insulation instead of average breakdown values. There are several methods to analyze a given set of data. Reference [38] shows that statistical distribution based on the 2-parameter Weibull model fits the best for mineral-based dielectric fluids. IEEE Statistical Committee [38] has classified Weibull data distribution to ascertain the reliability of a set of breakdown data of an insulation system. The Equations (4) and (5) below provide the 2-parameter model of Weibull distribution [38]:(4)$$F\left(x\right)=1-\exp{\left[1-\left(\frac{x}{\alpha }\right)\right]}^{\beta }$$
where F(x) = Probability of failure at a voltage
α = Scale parameter (=63.2% of data)β = Shape parameterx = Breakdown voltage

The experimental data is arranged on the probability paper by ranking it according to size and assigning a cumulative probability of failure (*P_F_*) to each data point given as:(5)$$P_{F}\left(i,n\right)=\frac{i-0.3}{n+0.4}\;100\%$$
where *i* = the *i*th data point
*n* = total number of data points

This data shall follow a reasonably straight line. In the present investigation, the analysis of data is based on the 2-parameter model of Weibull distribution, where *n* = 30.

Figure 8 compares the cumulative probability distribution of failure of data obtained for three types of oil samples, i.e., sresh mineral oil (MO), the nanofluid prepared using HMDS-coated silica NPs (MO-N), and nanofluid prepared using plasma-treated silica NPs (MO-NP). These distributions follow a linear relationship. The value of scattering of data (β) in each case is less than 7%.

Table 3 summarizes the AC breakdown voltage values at different failure probabilities (%P) for the three different fluids. It can be seen that there is a remarkable increase in the α breakdown values of nanofluid MO-N as compared to fresh mineral oil. This increase is +8% but with plasma treatment, this value increased to +23%. A similar trend is noted even at a 50% breakdown probability. The most interesting is the case at lower levels of breakdown probability of failure at 5%, which can be used for design purposes. It shows a 34% increase in nanofluid (MO-N) as compared to fresh oil, but the increase goes up to 60% for plasma-treated nanofluid (MO-NP).

Jin et al [27] have argued that the surface of silica NPs they employed in their experiments was hydrophilic so it could bind water dispersed in the oil on the surface of NPs making the surface more conductive and reported an increase in the breakdown of NF prepared using these silica NPs. Several other investigators have considered the influence of charge transport’s conduct in NFs from various aspects. Space charge is generated in a dielectric material under the action of high electric fields. When the rate of its accumulation is different from the rate of its dissipation in the electrode gap, its kinetic modifies the electric field which may cause either degradation or improvement in the insulation. The mechanism of space charge is therefore considered as the most prominent factor in determining the overall dielectric performance of the insulation system.

In the literature, charge trapping properties of NFs have been reported using a thermally stimulated depolarization current (TSDC) technique, whereas the sophisticated Pulse Electroacoustic (PEA) method is applied for the study of charge distribution in the stressed dielectric. Using these techniques, Du et al. [29] showed that both the shallow trap density and the charge dissipation velocity in NFs prepared using TiO_2_ NP greatly increased compared to those in pure MO, causing a significant increase in the breakdown performance of NF. They attributed it to the electron-hopping transport in a delocalized state. The fast electrons created due to the high field could be captured by the traps in the fluid and transported away by the hopping model. The fast electrons created under a high field are converted to slower electrons in the space charge traps and transported in the oil [39].

Similarly, Yuzhen et al [40] have reported the role of NPs in aged transformer oil and found an increase in AC breakdown voltage. They attributed the increase to the space charge distribution in the oil gap, the decay rate of space charge was found significantly increased due to shallow trap energy level (<0.8 eV) and higher trap density was induced, offering its rapid dissipation [41]. More recently, Rafiq et al. [42] also reported similar results by adding Al_2_O_3_ NPs in transformer oil. In brevity, all these experimental pieces of evidence and by several others reported in the literature, indicate that the shallow trap trappings de-trapping of electrons and higher charge dissipation rate causes an increase in dielectric properties. In contrast, if in an NF the space charge accumulation in the bulk of the oil is captured in deep trap centers with higher density and energy levels (≥1.0 eV) [41], the flow of electrons and their de-trapping rate is retarded while the mobility of ionized ions is slowed down. Such a space charge badly distorts the electrical field distribution in the oil and once its magnitude exceeds the threshold value, the electrical breakdown ensues at a much lower level than clean systems. We therefore strongly believe that with the addition of HMDS-grafted nano-silica particles, the (MO-N) sample undergoes a shallow trap mechanism, as illustrated above, which leads to increased breakdown voltage levels in the (MO-N) sample as compared to the clean MO sample [27]. The plasma treatment of NPs changes their surface chemistry, as illustrated earlier in Section 3.1. In particular, the (O-H) group attached to their surface contributes to increasing the number of hydrogen bonds (H^+^). The active hydrogen in hydrogen bonds can form hydrogen bonds with the -CH_3_ moieties and functionalities [43,44], which may lead to enhanced interfacial interactions of NP surfaces with the oil molecules, leading not only to a significant increase in the breakdown voltage of (MO-NP) nanofluid but also in its much-enhanced dispersibility and stability in transformer oil, as shown later in Section 3.5.

### 3.4. Role of Amino-Based Silane Coated Fumed Silica NPS

In the past few years, several investigators involved in European Union-funded “Horizon-2020” research project have exerted efforts to formulate a nano-polymeric composite that could provide the enhanced dielectric performance of HVDC cables for application in Ultra High Voltage (UHV) DC power networks. In this context, eight types of silanes were coated on fumed silica NPs and their impact was studied [25]. These silanes were divided into three groups: (i) non-polar (e.g., HMDS), (ii) polar silanes with amino moieties on their structure and, (iii) hydrocarbon-based silanes. The results showed that:
(a)Silica modified with unpolar silanes was easier to disperse in composite, while the ones modified with polar silanes agglomerated in the polymer matrix.(b)The nanocomposites filled with unpolar silanes increased the space-charge trap density, while the polar silanes caused an increase in charge trap depth and showed less injected charge, thus causing suppression of space-charge accumulation by introducing N_2_-rich polar functional groups. Thus, according to these results, amino-modified fumed silica is a promising candidate for polymeric nanocomposite for HVDC cable and capacitors applications.

Following the same strategy, Jin et al. [27] used silica nanoparticles and surfactant Z6011 silane to coat them. This is a coupling agent constituting aminopropyltriethoxysilane (APTES). This is a polar chemical with nitrogen atoms as an integral part of the surfactant. However, contrary to the favorable results obtained with amino-silane modified nano-polymeric composites, the addition of such NPs in insulating oil resulted in an almost 39% decrease in its AC breakdown strength [27]. This shows that the coating of NPs with silanes that have amino polar end moieties, as compared to HMDS (which is nonpolar), produces a negative impact on the space charge generated in the electrode gap filled with dielectric fluids. There are several reasons for this negative effect. Studies in polymers show that polar silanes such as amino groups produce deeper traps and higher trap density [6,26,41]. These trap charges also influence deeper trap energy (1.01 eV) and the trapping–de-trapping process [42].

The application of amino silane-grafted NPs in various dielectric fluids has also been investigated by some researchers [45,46]. The presence of APTES on NPs present in the oil is found to accelerate its thermal aging [45]. Bagwe et al. [46] showed that the introduction of amino silane-grafted NPs in fluids increases their aggregation, increasing their size and slowing mobility of space charge. Russ and Werner [47] have shown that polar compounds in transformer oil have been a major concern from the very beginning of the transformer industry. Polar compounds in just ppm levels can have detrimental effects on transformer oil due to aging and oxidation, vary by the chemistry of oil and are also responsible for their premature failures. All of the above findings suggest that amino-based silanes are not suitable for transformer oil applications as their presence will lead to the formation of deep traps and trap densities, which due to aggregation character exhibited in dielectric liquids, will render the ionic space change so formed less mobile. This will result in enhancement of the electrical field at the high voltage electrode that in turn shall decrease the dielectric strength of the oil.

### 3.5. Shelf Life of MO-NP Oil Sample

Figure 9 depicts digital images of oil samples, each prepared with 0.03 g/L loading of (a) silica nano-powder, and (b) nanofluid prepared using plasma-treated HMDS-coated NPs. It is clear that whereas sample (a) precipitated in just a couple of days, sample (b) was without any trace of sedimentation at the bottom, even after three months of shelf life. These results, therefore, demonstrate that the simple atmospheric pressure plasma treatment of fumed silica nanoparticles is a facile technique that effectively improved the dielectric stability of this NF. Use of this fluid in high voltage apparatus shall certainly reduce their size and maintenance cost.

## 4. Conclusions

HMDS-grafted fumed silica nanoparticles were used in this investigation. It is demonstrated that cold atmospheric-pressure plasma treatment of these NPs changed their surface chemistry, producing stronger chemical bonds and reducing weaker bonds. These resulted in excellent dispersibility and stability of the oil-based suspension. As a result, a significant improvement in dielectric strength was achieved. At a 5% probability of failure, this modified nanofluid exhibited a 60% increase in breakdown voltage, which is attributed to favorable space-charge trapping and de-trapping velocity in the oil.

The relative increase in conductivity of oil as compared to base oil is due to EDL. The relative permittivity decreased with temperature but was not much different from base oil in the temperature range up to 70 °C. Although plasma treatment of HMDS-treated silica increased the dielectric properties of the oil, their treatment with amino-polar silane considerably reduces their dielectric integrity. Therefore, in future studies, instead of polar silanes, the dielectric fluids should be treated with aliphatic silanes associated with different members of the alkoxy group.

## Figures and Tables

**Figure 1 materials-14-07649-f001:**
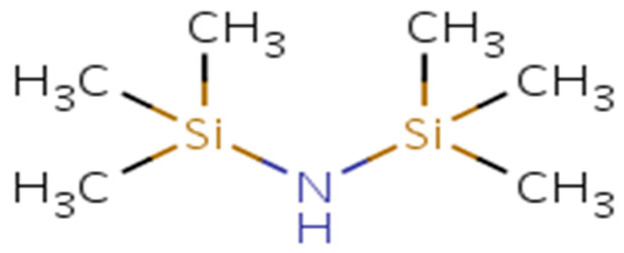
Chemical structure of HMDS.

**Figure 2 materials-14-07649-f002:**
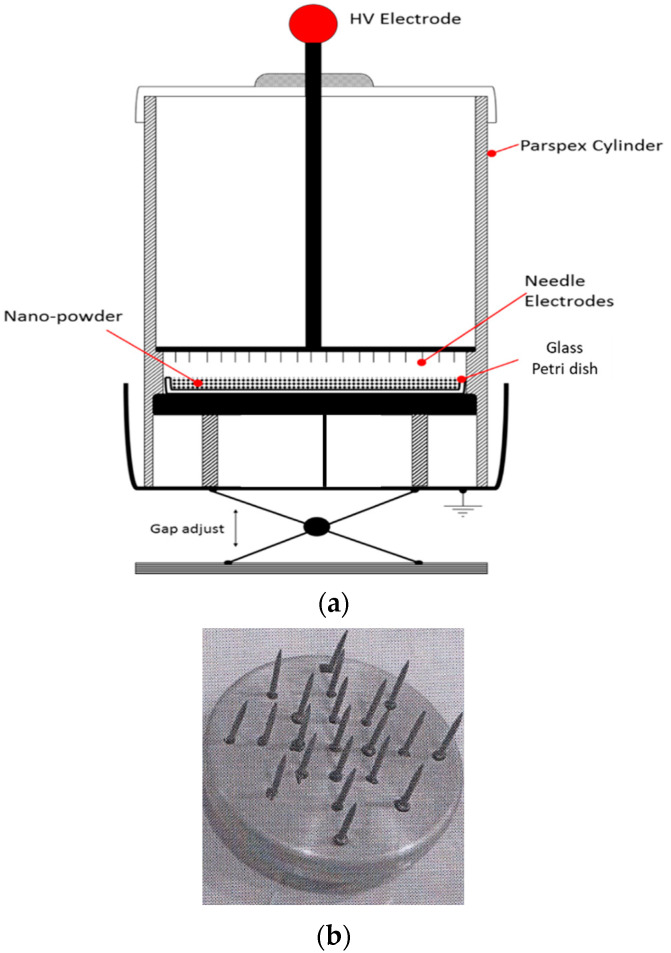
(**a**) Designed outline of Plasma reactor (**b**) High voltage electrode with an array of embedded steel needles.

**Figure 3 materials-14-07649-f003:**
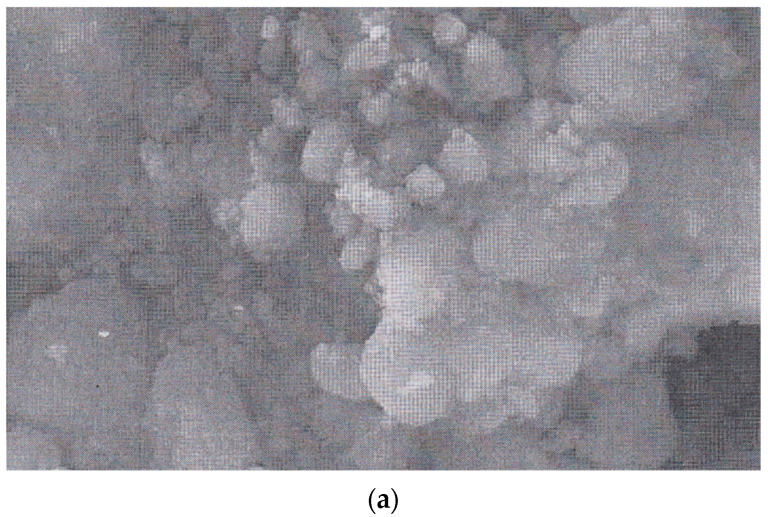
(**a**) Control sample of unmodified fumed silica NPs (**b**) Plasma treated NPs.

**Figure 4 materials-14-07649-f004:**
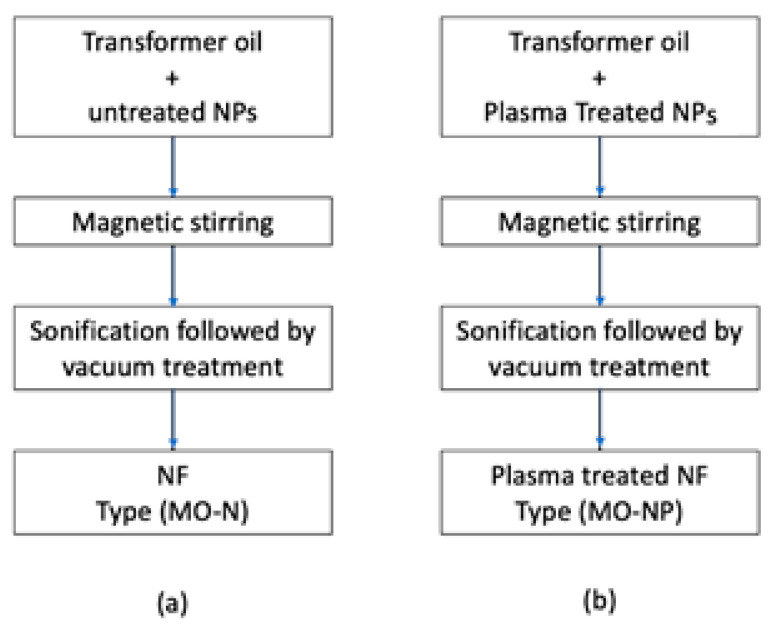
Flow chart for preparation of nanofluids with (**a**) fumed silica NPs (**b**) with plasma-treated fumed silica NPs.

**Figure 5 materials-14-07649-f005:**
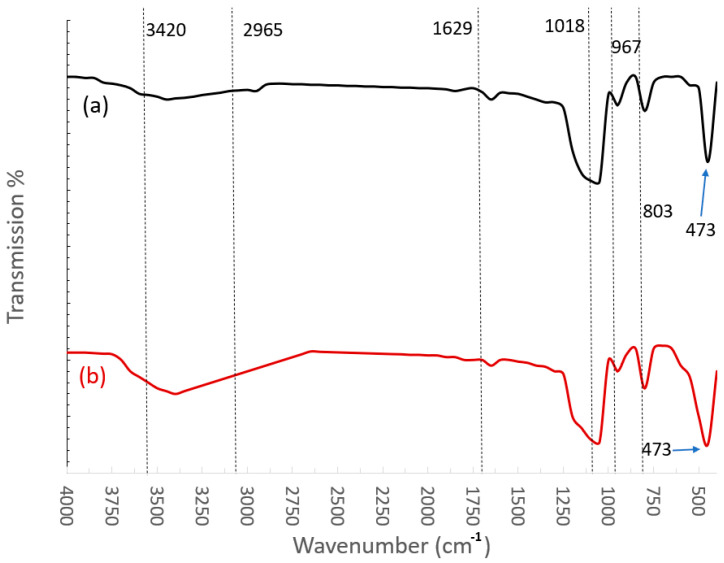
FT-IR spectrum of (**a**) HMDS capped nano-silica (**b**) After Plasma treatment.

**Figure 6 materials-14-07649-f006:**
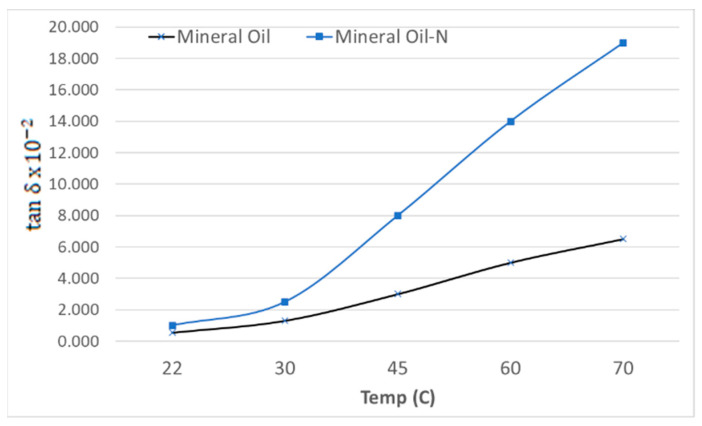
Variation of tan δ as a function of temperature.

**Figure 7 materials-14-07649-f007:**
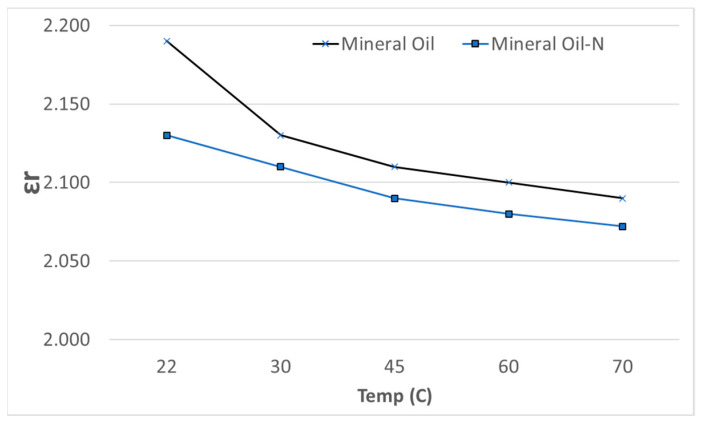
Variation of $\varepsilon_{r}$ as a function of temperature.

**Figure 8 materials-14-07649-f008:**
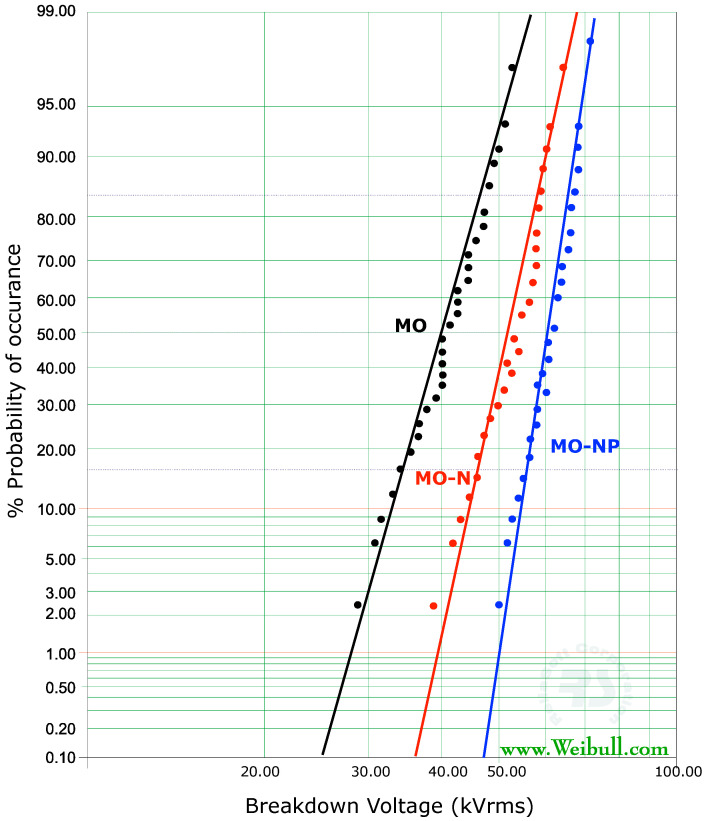
Comparison of the cumulative probability distribution of breakdown voltage in three oil samples.

**Figure 9 materials-14-07649-f009:**
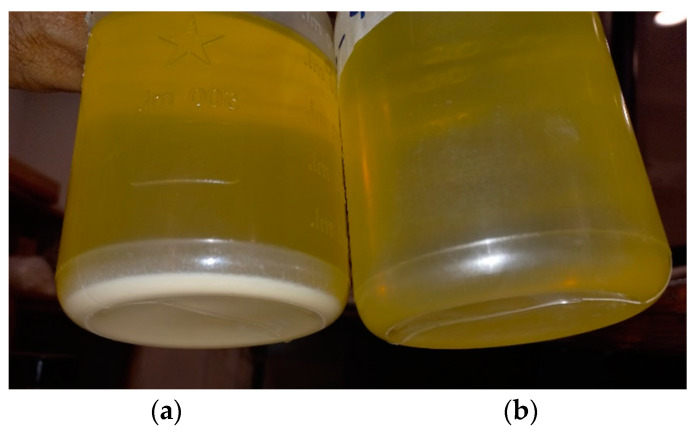
Digital image of stored nanofluids (**a**) NF with untreated SiO2 (**b**) NF with plasma-treated HMDS-coated nanoparticles.

**Table 1 materials-14-07649-t001:** Characteristics of hydrophobic fumed silica.

Characteristic	Value
**pH**	7–7.5
**Moisture Content**	<0.5%
**Color**	White
**Surface Area**	260 m^2^/g
**Avg. Particle size**	7 nm
**Composition**	**wt %**
**SiO_2_**	99.8
**Al_2_O_3_**	≥0.05
**Fe_2_O_3_**	≥0.01
**TiO_2_**	≥0.03

**Table 2 materials-14-07649-t002:** Change in atomic concentration of O, Si, and ratio (O/Si) on the surface of untreated and plasma-treated silica NPs.

	Atomic Content (%)		
Sample	Si	O	(O/Si) Ratio
**Untreated**	24	75	3.12
**Plasma Treated**	32	68	2.12

**Table 3 materials-14-07649-t003:** Breakdown voltage ($kV_{rms}$) of oil samples as a function of %P.

%P	MO	MO-N	% Increase	MO-NP	% Increase
5	32	43	+34	51	+60
50	42	53	+24	62	+47
63.23	52	56	+8	64	+23

## Data Availability

Not applicable.

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
