# Peer review of "Probing the Use of Silane-Grafted Fumed Silica Nanoparticles to Produce Stable Transformer Oil-Based Nanofluids"

_materials, 2021, doi:10.3390/ma14247649_

Round 1
Reviewer 1 Report
The manuscript reports on the production and characterization (morphological, compositional, and electrical) of mineral oil-based fluids containing surface-treated silica nanoparticles in suspension. The fabricated nanofluids are addressed to work as insulating oil in electrical transformers. The data included in the submitted version of the manuscript are interesting and potentially can attract the attention of the Journal’s readership. However, the manuscript presents gaps in regard to its presentation, literature review and key morphological information. Therefore, to the best of my understanding, the authors should upload a revised version of the manuscript, either incorporating or rebutting (one by one) the comments listed below.
- The manuscript’s technical English should be revised from the abstract down to the conclusions. In this regard, special attention should be paid in the following points, not exhausting in the presented list. Chemical nomenclature states that chemical elements and compounds should be written in lower case instead of starting with upper case. On page-2, line-50, instead of “singular and multi-nanoparticle”, it would be more appropriate writing “single- and multi-nanoparticle”. On page-2, line-54, instead of “matter”, it would be more appropriate writing “issue”. On page-2, line-90, instead of “oil matrix”, it would be more appropriate writing “oil-based suspension”. On page-2, line-96, instead of “ranges”, it would be more appropriate writing “range”. On page-3, line-105, instead of “while”, it would be more appropriate writing “whereas”. Similar mistake can be found in different spots along the manuscript and should be double checked as well. The representation of the volume unit appears in the manuscript (many spots) as “ml” and should be corrected by “mL”. On page-3, line-132, the authors stated “are presented elsewhere”, but didn’t provide any reference. Reference(s) should be included at the end of this sentence and the reference list fixed accordingly. On page-5, line-161, instead of “were”, it would be more appropriate writing “was”. On page-6, panel-(b) of Figure 4, instead of “NP3”, please write “NPs”. On page-6, the subtitle in 2.4. is the same as the subtitle in 2.3. Please, check and fix it accordingly. On page-6, line-192, instead of “MO oil”, it would be more appropriate writing simply “MO”. On page-7, line-208, instead of “call”, it would be more appropriate writing “cell”. On page-7, line-217, please double check the wavenumber 1625 with the wavenumber included in Figure 5. Likewise, double check the wavenumber 1108 included in line-219 with the wavenumber included in Figure 5. On page-7, line-218, please double check the “silinol group” nomenclature against “silanol group”. On page-7, line-229, instead of “group”, it would be more appropriate writing “band”. On page-12, line-405, instead of “apolar”, it would be more appropriate writing “nonpolar”.
- On page-2, line-66, the molecule of argon (noble gas) is represented as diatomic (Ar2) and should be corrected to monatomic (Ar). On page-2, line-80, the authors introduce the hexa-dimethylsiloxane (HMDS). The authors are requested to double check the nomenclature of this compound. Note that hexamethyldisiloxane (HMDSO) corresponds to the chemical structure presented in Figure 1. The same acronym appears in different spots along the manuscript and should be double checked as well, as for instance in the caption of Figure 1.
- On page-4, line-144, the authors stated “With the naked eye”. However, the micrographs presented in Figure 3 show scale bars of 100 microns. In addition, typical sizes of nanoparticles in both micrographs are smaller than 100 microns, though visibly in the size range of microns. Then, how come the authors can state that “With the naked eye, no change in the morphology of NPs was observed”? Moreover, a couple of lines below in the text (lines 149-150) the authors made a contradictory statement (“…break weaker bonds in the silica aggregates during the plasma modification which resulted in smaller dimensions of the plasma modified silica units.”). Surprisingly, Table 1 collects the characteristics of the hydrophobic silica and includes the average particle size of 7 nm. In view of the apparent conflicting information and for the best information to the journal’s readership, this reviewer is requesting the authors to add the particle size histograms of the samples whose SEM micrographs appear in Figure 3. The histograms could be positioned in the right hand-side of each SEM micrograph. The fittings of the particle size histograms using a chosen distribution function should be included as well. While preparing the particle size histogram, please be aware of using the partition criteria proposed by Sturges (1926), relating the total number of Classes (C) with the number (N) of assessed particles via C=1+3.322*log(N). The authors can check the use of the Sturges’ relation in the paper by Aragon et al. in J. Phys. Cond. Matter vol. 27, paper 095301, 2015 (DOI: 10.1088/0953-8984/27/9/095301). The authors should consider the inclusion of this paper in the revised reference list.
- In the Introduction section the authors correctly stated that “These NFs have gained the attention of the researchers worldwide due to the enhancement of their insulating properties and much better cooling performance.” However, the authors devoted the focus of the Introduction section to the insulating characteristics of the transformer oil, leaving the text completely unbalanced in regard to the key issue of cooling performance. In this regard, the use of engineered magnetic nanoparticles plays an important role, enhancing the cooling performance while simultaneously improving the insulating characteristics. Interesting, de Almeida et al. (J. Alloy Compd. vol. 500, pp. 149-152, 2010 (DOI: 10.1016/j.jallcom.2009.10.245) reported successful fabrication of core-shell maghemite-silica nanoparticles that incorporates two key material characteristics for preparation of highly efficient nanofluids addressing electric transformers. While revising the Introduction section to make it more balanced, the authors should consider the inclusion of the references mentioned above.
Indeed, I found the manuscript interesting but there are points to be revised, as indicated above, in terms of presentation, balanced content and reference list, that need to be bridged before this reviewer can support the manuscript publication in Materials journal.
Author Response
Comment 1:
We are extremely thankful to the reviewer for a very critical and thorough review of our paper and for the corrections and several valuable suggestions extended.
- The technical English of the manuscript now stands corrected. We have employed the “Premium Grammarly Online tool” to fix grammatical errors. We are glad to report that ALL errors have been fixed.
- Chemical nomenclatures are corrected. All typo errors have been corrected.
- Text and errors in FTIR wave- numbers related to figure 5 now stand corrected.
Comment 2:
Figure 1 is replaced with a correct chemical structure of HMDS.
Comment 3:
The word “naked-eye” was actually used with the intention that no discoloration or any other visible change occurred in nanoparticles after plasma treatment. Perhaps it was not properly phrased and caused confusion. However, the corresponding text is now modified and corrected.
Your concern about the scale- bars of 100 microns are valid. Our SEM technician has told us that due to ingress of a bug in its software, the scale- bars are not showing correctly and the problem is being looked into. To avoid confusion, we have cropped the scale bars in the SEM micrographs of figure 3 (a, b). The purpose of comparison was to distinctly exhibit the images of nanoparticles before and after plasma treatment. Plasma generated in a dielectric barrier discharge (DBD) setup working under atmospheric pressure constitutes of either “filamentary discharges” or “diffused discharge”. The filamentary discharges (also referred to as “micro-discharges”) occur due to electrical breakdown in small air-gap constituting of high voltage electrode and a dielectric barrier (quartz glass in our setup) occur in a large number of short-duration bright filaments which impinge at the glass barrier. So in our designed plasma reactor, filamentary plasma pulses are generated. It is clear in figure 3 that untreated nanoparticles are in the form of agglomerates, whereas the plasma treatment has loosened them, and individual nanoparticles are showing somewhat spherical structures. This is due to the result of impinging micro-discharges on the surface of nanoparticles, changing their surface chemistry as illustrated later in section 3.1.
The particle size and their distributions can be accurately measured either through TEM or using the Dynamic Light Scattering technique (i.e. photon Correlation Spectrometry). Unfortunately, we do not have these facilities at our premises and are unable to add this information.
Comment 4:
The role of cooling performance of nanofluids has been documented in a dedicated paragraph of the Introduction, which includes the contribution of de Almeida et al. (see reference [5] in the list of references).
Reviewer 2 Report
The authors have experimentally investigated the hydrophobic silane grafted fumed nano-silica
employed in transformer oil to formulate nanofluids. The study is relevant and interesting. The
subject of the study is about nanofluid excellent use. Compared to other available published work, the
authers work shows improved results. The manuscript is written in systematic and clear way. The
study addresses the main question posed and have summarised the results in conclusions. Minor
revision is suggested for the manuscript.
(1) After Choi et al. [1], there have been made latest innovations regarding nanofluids so the
authors should read and cite the following relevant papers.
https://doi.org/10.3390/app7030271, http://dx.doi.org/10.1016/j.rinp.2017.10.017
(2) It would be better to use color photos/figures in Figure 3 (a), (b).
(3) Provide the nomenclature.
(4) Provide the comparison with published work.
Thanks
Author Response
Many thanks for your valuable comments and critical review.
Comment 1:
A dedicated paragraph has been added in Introduction that describes the role of thermal efficiency of nanofluids including latest innovations. Your suggested reference is incorporated here as ref. [11].
Comment 2:
We are sorry, our SEM software does not have option for colored images.
Comment 3:
Instead of nomenclature, we have defined all symbols and abbreviations, wherever they appear in the text.
Comment 4:
Comparisons are incorporated wherever they are required.
Reviewer 3 Report
The present paper is an original article about hydrophobic silane grafted fumed nano-silica used in transformer oil to synthesis of nanofluids. It was used cold atmospheric pressure plasma treatment of nanoparticles changing the surface chemistry by producing stronger chemical bonds and reducing weaker bonds.
The nanofluid sample having plasma-treated nanoparticles in composition showed excellent dispersibility and stability of the oil matrix. In addition, a significant improvement in dielectric strength and conductivity increase of oil were achieved.
It is an important work - that could be helpful to researchers, regarding the producing of stable nanofluids based on transformer oil with improved properties.
The paper is well organized and easy readable.
I recommend accept the paper after minor revision.
Observations for revision:
Line 38 - AC-BDV abbreviation might be write here - being the first time when you used that term in manuscript.
Line 168 and line 184 - you used the same for the both subchapters: "Nanofluid preparation"....please correct this situation or combine both subchapters..
Line 265 - please specify the abbreviated terms for EDL from the first using for better understanding by any type of readers.
Line 413 - Replace “Bagawe et al.” with “Bagwe et al.” (reference 33)
Line 415 and line 538 (reference 34) are not in concordance ... please revise!
Author Response
Thanks for suggesting several typo errors and missing details of abbreviations. These and a few others pointed out now stand rectified.
Reviewer 4 Report
The authors failed to describe the importance of Figure 8. They should add a discussion in the section.
Author Response
Thanks for the critical review of our paper.
Comment 1:
Section 3.3 adequately illustrates the purpose and importance of figure 8.
Reviewer 5 Report
The paper has good work done for probing the use of silane grafted fumed silica nano particles to produce stable transformer oil-based nanofluids but there are some items should be followed to be available for publishing in MPDI-Materials journal as follows:
- Review the English language with native reviewers.
- Introduction is too long, it should be concise with recent refs.
- The novelty doesn’t appear in the paper, there is no presentation of novelty was elaborated in the paper.
- Try to compute and analyze electric breakdown with respect to recent references eg.:
- Thabet, M. Allam and S.A Shaaban, “Assessment of Individual and Multiple Nanoparticles on Electric Insulation of Power Transformers Nanofluids” Electric Power Components and Systems Journal, Vol. 47, Issue 4-5, pp. 420-430, 2019.
- Concise the conclusion with the scientific new points addition(s).
- To discuss new idea, it is more efficient to add a comparative study with:
- Variant nanoparticles percentages inside the nanofluids.
- Recent references “breakdown, tand δ, ….etc”.
- Preparation nanofluid procedures.
Author Response
Thank you for your critical review.
Comment 1:
English text issues have been resolved.
Comment 2:
Introduction is modified as per the suggestions of reviewers, while other sections are also corrected accordingly.
Comment 3:
Our paper is experimental based and deals with single nanoparticles grafted with silane moieties. The subject matter given in the reference “Thabith et al. …Assessment of Individual and Multiple Nanoparticles”…., does not match with the subject matter treated in this paper and comparing it with present work is therefore regretted.
Reviewer 6 Report
I have attached a doc version of the paper with my comments. The paper sounds interesting, however:
- Some corrections are required to improve the overall layout of the figures/graphs
- Numerous references are missing - many statements are not supported nor claimed to be original input
- Some corrections to research results must be done.

Author Response
We are extremely thankful to the reviewer for a very critical and thorough review of our paper and for the corrections and several valuable suggestions extended.
KO1:
It is a universally well-known fact among the electrical engineering community and doesn’t need any typical reference.
KO2:
These are already present in the modified introduction.
KO3, KO4:
It is also well known among the manufacturers of equipment and power utilities. See ref. [2].
KO5:
Corrected.
KO6:
Explanation included in the current paragraph.
KO7:
This Abbreviation is already explained in the abstract.
KO8:
We use the common convention of listing the abbreviations in full the first time as they appear in the manuscript. For any later use, and as recommended by the other reviewers, we have listed only the abbreviation.
.
KO9, KO10:
Explained already in the revised text.
KO11:
This is indeed a new area of research. We have referred to this paper in the introduction as Ref.[46] This fluid will be considered for future preparation of ionic nanofluids.
KO12:
Corrected.
KO13:
It is well known that insulating oils used in transformers and for high voltage applications are mineral-based, derived out of petroleum crude distillates.
KO14-KO17:
The purity of fumed silica is 99.8%. please refer to Table 1 for more detailed information.
KO18-KO20:
All corrected.
KO21:
A very hurtful remark on your part. This is a tailor-made plasma reactor designed and implemented at our premises.
KO22-KO25:
Details of the sonicator are now included in the text.
KO27:
For Si-O-Si we have ref. [47] added to the list of references.
KO28-KO30:
Unfortunately due to the limitations, we cannot perform it with our FTIR software.
KO31:
The Y-axis was not shown in Figure 5, we have added it.
KO32:
We have seen and compared the expanded view generated by the software, but not shown here separately.
KO35, KO36:
Electrical power engineers are well versed with this effect. They want to keep a regular check on their assets. This is an important property that is examined on a periodic basis. The power frequency (50Hz) is used because the electrical power system operates at this frequency. However, to check the insulation, tests are also performed in the frequency range of 0.1Hz to 1000Hz. This technique is called Dielectric Spectroscopy and is mostly used to assess the condition of high voltage cables.
KO37:
Corrected.
KO38:
We used ÌŠC since power engineers commonly use this unit of temperature.
KO39:
Please refer to details given in ref. [35]
KO40-KO42:
We regret to include this request since similar figures have been plotted and presented in several research papers and are well accepted by the scientific community.
KO43:
Regarding the missing uncertainties, the evaluated data of all three different graphs shows less than 7% dispersion. This is commented in the text before Fig. 8.
Round 2
Reviewer 1 Report
The present version of the manuscript incorporates almost all suggestions I have made in my previous report. However, before this manuscript can be accepted for publications the authors NEED to check the reference ordering in the main text, particularly in the Introduction. The sequence of the references in the main text is simply messy.
Author Response
Response to Reviewers
We thank the reviewer for helping us improve this manuscript. We have updated the ordering of references as they appear (first) in the text. All the reference citations in the text are marked with yellow color.
Previous |
New |
1,2,3,4 |
1,2,3,4 |
5 |
9 |
6 |
11 |
7 |
12 |
8 |
13 |
9 |
14 |
10 |
15 |
11 |
16 |
12 |
17 |
13 |
18 |
14 |
5 |
15 |
19 |
16 |
20 |
17 |
21 |
18 |
22 |
19 |
23 |
20 |
24 |
21 |
25 |
22 |
6 |
23 |
26 |
24 |
27 |
25 |
28 |
26 |
29 |
27 |
30 |
28 |
31 |
29 |
32 |
30 |
7 |
31 |
8 |
32 |
34 |
33 |
35 |
34 |
36 |
35 |
37 |
36 |
38 |
37 |
39 |
38 |
40 |
39 |
41 |
40 |
42 |
41 |
43 |
42 |
44 |
43 |
45 |
44 |
46 |
45 |
47 |